# Orchid species diversity across a forest disturbance gradient in west Mau forest, Kenya

Job N. Mirioba(ORCID)[1], William Emitaro[1], Benson Obwanga[2], Humphrey Gaya[3], Nereoh Leley[4], John Otuoma[5], John M. Maina[1], Fanuel Kawaka(ORCID)[1]*

1 Jaramogi Oginga Odinga University of Science and Technology (JOOUST), Bondo, Kenya, 2 Laikipia University (LU), Nyahururu, Kenya, 3 Genetic Resources Research Institute (GeRRI), Kikuyu, Kenya, 4 Kenya Forestry Research Institute (KEFRI), Londiani, Kenya, 5 Kenya Forestry Research Institute (KEFRI), Kisumu, Kenya

* fkjairo@hotmail.com

**Data Availability Statement:** All relevant data are within the manuscript and its Supporting Information files.

## Abstract

Orchidaceae is one of the most diverse and widespread groups of flowering plants. Despite their immense ecological and socio-economic value, their spatial distribution across forest disturbance gradient is not well understood, particularly in tropical montane forests. This study assessed the influence of forest degradation on orchid species richness and abundance in West Mau Forest, Kenya. Stratified systematic sampling was adopted across three different disturbance regimes consisting of relatively intact forest, moderately disturbed forest and highly degraded forest. A total of five orchid species were recorded from nine host-tree species. The intact forest had seven host tree species with five orchid species. The moderately degraded forest had four host-tree species with two orchid species, while the highly degraded forest that had no orchids. *Polystachya confusa* was the most abundant orchid species (600.0±227.9 clumps ha$^{-1}$) followed by *Bulbophyllum* sp (340.0±112.2 clumps ha$^{-1}$), *Chamaeangis* sp (300.0±115.5 clumps ha$^{-1}$), *Aerangis* sp (200.0±57.7 clumps ha$^{-1}$) and *Tridactyle* sp (100.0±0.0 clumps ha$^{-1}$). The results of this study indicate that forest degradation reduces orchid species diversity in tropical montane forests. They also show that orchids are bioindicators of forest degradation status.

## 1. Background

The family Orchidaceae forms the most diverse and widespread group of flowering plants [1]. Its plants have unique ecological and socio-economic value [2]. Apart from being bioindicators of ecosystem integrity, they are of great medicinal [3], and ornamental and floricultural value [4]. Many orchids are currently produced for commercial purposes as cut flowers because of their diverse colour range, which varies from deep purple to brilliant blue, vibrant orange, potent pink, beautiful white and delightful red [5]. Although certain orchid species are found in temperate and boreal zones, the majority are indigenous to tropical regions [6]. In Africa, there are more than 3,545 species of orchids, with Central Africa being the richest with

**Funding:** The work was supported by a grant from the Rufford Foundation (No. 36278-C).

**Competing interests:** The authors have declared that no competing interests exist.

708 orchid species [7]. Kenya has approximately 283 orchid species, which belong to 50 genera, and half of these are epiphytic [8, 9].

Since tropical orchids predominantly occupy forest ecosystems, the reduction in closed-canopy forest cover in Kenya from about 12% to under 2% over the past four decades is feared to have subjected these plants to the danger of extinction [10]. It is hard, however, to predict the extent and severity of such danger because the diversity and distribution of these plants remain unstudied in most forest ecosystems in Kenya [11]. Moreover, the effect of loss of tree species from closed-canopy forests in Kenya on the population of epiphytic orchids remains largely unknown [12]. Recent orchid population surveys in Mt. Elgon Forest National Park [13] and West Mau Forest in Kenya [8] have recommended a comprehensive assessment of threats to orchid populations with regard to forest degradation and the effect on orchid species diversity and distribution. Baasanmunkh, Oyuntsetseg [14] identified six anthropogenic threats to wild orchid populations, namely: agricultural expansion, human settlement, the introduction of exotic tree species in their habitats, overgrazing, over-exploitation due to high orchid market demand and traditional use as herbal medicine. On the other hand, Evans, Kamweya [15] identified degradation of forest ecosystems as the greatest threat to orchid populations. An earlier study [16] examined the threat presented by forest degradation to the population of wild orchids in the Mau Forest Complex. However, the study did not clearly illustrate the effect of forest degradation on the diversity and spatial distribution of wild orchid populations.

Since epiphytic orchids tend to be host-specific, their populations are likely limited to a few phorophytic species (orchid-bearing trees) in a forest ecosystem [17, 18]. Epiphytic orchid species diversity and abundance tend to be significantly greater in relatively undisturbed habitats [19, 20]. Similarly, taller trees tend to have higher wild orchid numbers due to exposure to light and higher humidity unlike shorter ones [21]. Thus, some studies propose that old-growth native tree species should be protected to preserve the populations of wild orchid species [22, 23]. Forest degradation and loss of host trees would lead to a decline in orchid biodiversity because their preferred habitats would have been destroyed [24]. Previous studies have explored the association between epiphytic orchids and phorophytic trees and described the relationship as commensalistic [25]. According to Ackerman [26], some orchid species grow at random in relation to phorophytic tree species, while others prefer specific tree species and avoid others. However, host-species preferences of wild orchids across forest disturbance regimes has not received much attention, specifically in montane forests. The current study assessed the variation in orchid species diversity across a forest disturbance gradient in West Mau Forest, a tropical montane forest ecosystem in Kenya.

## 2. Materials and methods

### 2.1 Study site

The study was carried out in West Mau Forest Block in Kericho County (Fig 1). It is one of the 22 blocks of the Mau Forest Complex in Kenya and covers 22,700 ha [27, 28]. It lies between latitude 0˚ 33' South and 35˚ 21' East at an elevation of 2,000 to 2,800 meters above sea level [29]. It experiences a cool and wet climate with an annual rainfall of 1,500 mm to 2,100 mm and a mean day temperature of 23˚C [30]. The geology of the area comprises quartzites and gneisses, which cover Western Mau and extend to South-Western Mau [31]. The soils of the area consist of deep, fertile volcanic soils [32]. The forest is rich in biodiversity with a high level of endemism for plant species [33]. Its key tree species include *Pouteria adolfi-friedericii*, *Strombosia scheffleri*, *Polyscias kikuyuensis*, *Olea capensis*, *Prunus africana*, *Albizia gummifera* and *Podocarpus latifolius*. The forest is of significant economic and ecological benefit to the

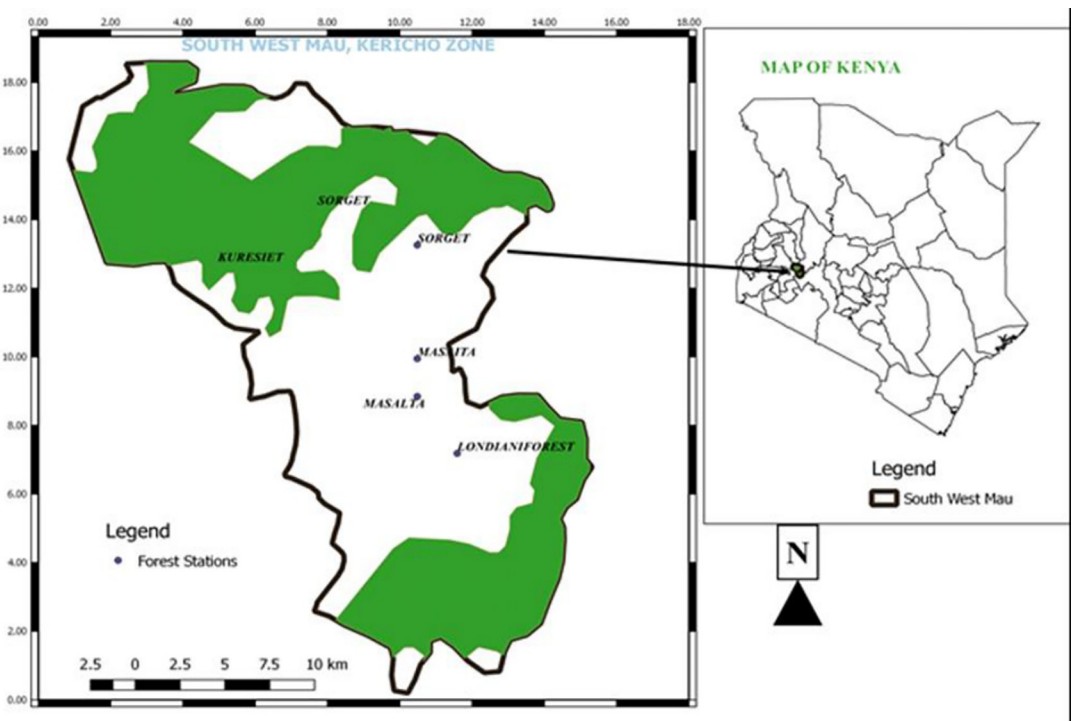

**Fig 1. South western Mau forest, Kenya.**

local community, the country and the global environment [34]. It is a major water tower that drains into Lake Victoria.

## 2.2 Study design

The study employed stratified systematic sampling based on three different forest disturbance regimes, namely: intact forest, moderately disturbed forest and highly degraded forest [35]. The three disturbance regimes were located within the same forest station, Kericho Forest Station. The classification and characteristics of the disturbance regimes (Table 1).

The intact site was located in Saosa Compartment and it comprised a mature, multistorey tropical forest stand that had not been subjected to commercial logging. The moderately disturbed site was situated in Cheboswa Compartment and it resulted from selective logging of the original intact forest. The highly degraded site was located in Kiburgen Compartment and its degradation was caused by the subsequent removal of remnant trees from the selectively

**Table 1. Description of different forest disturbance regimes.**

| Forest disturbance regime | Description |
|---|---|
| Relatively intact forest | A forest stand that has retained over 75% of its original tree cover. It may have been disturbed by selective removal of trees by local communities, but has never been subjected to commercial logging |
| Moderately disturbed forest | A forest stand of 30–50% tree cover as a result of selective removal of trees resulting in significant forest canopy gaps. Tree removal may have occurred through poaching by the local community or commercial logging. |
| Highly disturbed forest site | A field layer resulting from the clear-felling of a forest stand. The open fields are dominated by grass or scrub and remnant trees. |

logged forest site by the local community. The three compartments shared common boundaries. They had varying widths ranging from 100 m to about 300 m. Their lengths ranged between 1 km and 2 km. Prior to the logging operations, the study site was one single closed canopy tropical montane forest. The logging operation created a forest degradation gradient in which the severity of forest degradation increased with distance from the forest to the community boundary.

During this assessment, three sub-blocks were selected from each of the forest disturbance regimes. Assessment entailed the use of 0.5 km line transects laid along each sub-block in a forest disturbance regime. In order to avoid the edge effect caused by narrowness of some sub-blocks (100m to 150 m wide), only one transect was laid in each of the three sub-blocks under a forest disturbance regime. Five sample plots of 20 m × 5 m were located along a transect at intervals of 100 m. The sample plots were located using a Germin ETrex 32x GPS receiver. The 20 m × 5 m main plots were used to assess trees with a diameter at breast height (DBH) of at least 10 cm. Young trees and saplings (of <10 cm DBH) were assessed in 5 m × 5 m sub-plots, which were nested within the 20 m × 5 m plots. There was one sub-plot per sample plot. A sub-plot was located on the left-hand side corner of a sample plot in a northern orientation. Thus, three transects were laid in each forest disturbance regime, while each transect had ten sample plots. The total number of sample plots was therefore 90, comprising 45 main plots and 45 sub-plots.

## 2.3 Data collection

Data were collected in the 20 m × 5 m sample plots targeting trees of 10 cm DBH or more, and the 5 m × 5 m sub-plots for trees less than 10 cm in DBH. Both orchid host trees and non-host trees were recorded. Woody plant species found within the sample plots were recorded by name, DBH and crown height. The trees were recorded as phorophytic (orchid-bearing) or not. For phorophytic trees, the number of orchid clumps was recorded together with the height of each clump on the host tree. The number of individual orchid species were counted and recorded. A similar sampling procedure was applied to young trees and saplings with DBH < 10 cm in the 5 m × 5 m sub-plot. Details of the forest stand structure were captured for the purpose of understanding woody species distribution by canopy stratification. This included stem counts, stem DBH and canopy height of trees within the sample plots. Trees were distinguished as host plants or non-host plants to inform the influence of forest degradation status on the orchid species richness and abundance. Orchids were identified based on morphological features, such as the shape and colour of the flowers, stems and leaves, and number of leaves and flowers (Fig 2). In situations where field identification was not possible, orchid samples were photographed and sent to the laboratory for identification using existing specimens in the Flora of Tropical East Africa Species catalogue.

## 2.4 Data analysis

The data collected were entered into MS Excel spreadsheets for processing. Analysis entailed both descriptive and inferential statistics. Descriptive statistics was used to compute tree stem density (number of tree stems per ha), mean stem DBH, mean tree stem height, orchid-host tree species richness and abundance, mean stem DBH and height of host trees, orchid clumps per host tree and clump density (clumps her ha) for each forest disturbance regime. These variables were then subjected to analysis of variance (ANOVA) using Genstat 21st Edition to determine whether the differences observed from descriptive statistics were statistically significant. The Ryan-Einot-Gabriel-Welsch multiple range test was used to condut post-hoc tests to separate means.

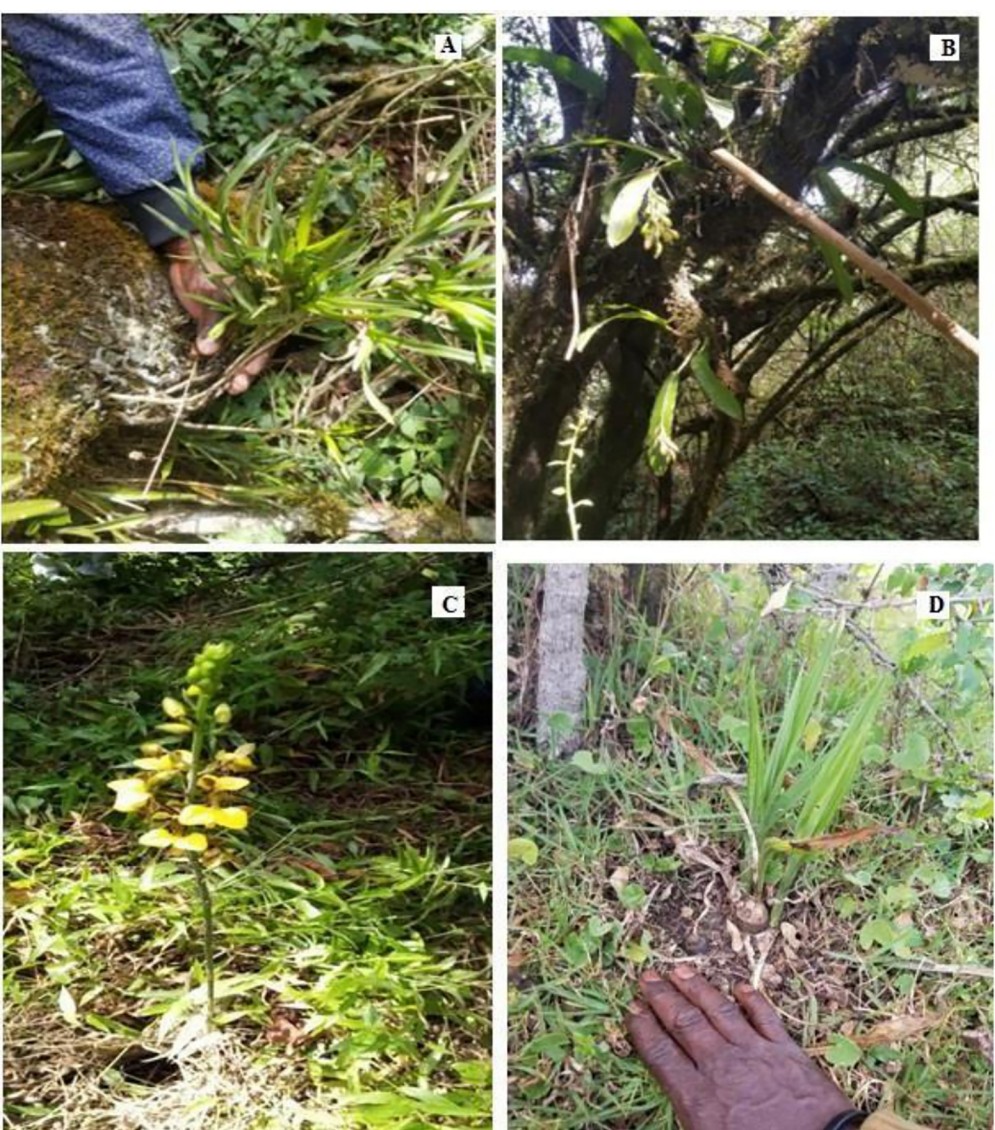

**Fig 2.** Photos: A (Aerangis sp) epiphytic orchid; B (Bulbophyllum sp) epiphytic orchid, C (Eulophia sp, with yellow flowers) terrestrial orchid, and D (Eulophia sp) terrestrial orchid.

## 3. Results

### 3.1 Host tree species richness and abundance

A total of 32 tree species were recorded by this study. Nine out of the 32 tree species hosted orchid species. These were *Albizia gummifera*, *Dovyalis macrocalyx*, *Macaranga kilimandscharica*, *Neoboutonia macrocalyx*, *Parvetta grandifolia*, *Podocarpus sp*, *Schefflera abyssinica*, *Syzygium guineense* and *Tabernaemontana stapfiana*. Seven host tree species were in the intact forest, while four were recorded in the moderately disturbed forest. The overall tree stem density was 472 woody stems ha$^{-1}$, of which orchid host trees accounted for 198 stems ha$^{-1}$ (41.9% of the woody plants population). Host tree species in the intact forest were generally taller ($F_{(1,2)} = 6.57$; $p = 0.062$) and larger in diameter ($F_{(1,2)} = 3.14$; $p = 0.151$) compared to those in the moderately disturbed forest (Table 2). The moderately disturbed forest had

**Table 2. Structural attributes of orchid host trees across a forest degradation gradient in West Mau Forest, Kenya.**

| Forest disturbance regime | Orchid host trees | | No. of orchid species | Orchid abundance (clumps ha⁻¹) |
|---|---|---|---|---|
| | Mean stem DBH (cm) | Mean tree height (m) | | |
| Intact forest | 46.6 | 24.6 | 5 | 390±133.8 |
| Moderately disturbed forest | 26.2 | 13.3 | 2 | 422.2±195.6 |
| Highly degraded forest | - | - | - | - |
| | $p = 0.151$ | $p = 0.062$ | | $p = 0.933$ |

relatively more orchid clumps per ha than the intact forest. There were no orchids in the highly degraded forest site.

None of the five orchid species belonged to a single host tree species. *Polystachya confusa* was present on eight different host tree species while the rest were recorded in three host trees each (Table 3). Of the nine host tree species, seven shared between two and four orchid species, while two had single orchid species. The number of orchid clumps per host tree ranged between 1 and 5.2, while the orchid clump height on host trees ranged between 1 m and 25 m above the ground.

## 3.2 Orchid species richness and abundance

A total of five orchid species were recorded across the three forest disturbance regimes. These were *Aerangis* sp, *Bulbophyllum* sp, *Chamaeangis* sp, *Polystachya confusa* and *Tridactyle* sp. Of the five species, *Polystachya confusa* had the highest population density (600.0±227.9 clumps per ha) compared to *Tridactyle* sp that recorded the lowest (100.0±0.0 clumps per ha) (Fig 3).

All the five orchid species were present in the intact forest. The moderately disturbed forest had only *Aerangis* sp and *Polystachya confusa*, while the highly degraded forest did not have

**Table 3. Orchid host tree species and their respective orchid clump densities in West Mau Forest, Kenya.**

| Host tree species | Orchid species | Orchid clumps per host tree |
|---|---|---|
| *Albizia gummifera* | *Chamaeangis sp* | 1 |
| | *Polystachya confusa* | 1 |
| | *Tridactyle sp* | 1 |
| *Dovyalis macrocalyx* | *Polystachya confusa* | 3.8 |
| *Macaranga kilimandscharica* | *Aerangis sp* | 1 |
| | *Chamaeangis sp* | 5 |
| | *Polystachya confusa* | 1 |
| | *Tridactyle sp* | 1 |
| *Neoboutonia macrocalyx* | *Aerangis sp* | 3 |
| | *Bulbophyllum sp* | 1 |
| | *Polystachya confusa* | 5.2 |
| *Parvetta grandifolia* | *Aerangis sp* | 2 |
| | *polystachya confusa* | 2.7 |
| *Podocarpus sp* | *Bulbophyllum sp* | 2 |
| *Schefflera abyssinica* | *Polystachya confusa* | 2 |
| *Syzygium guineense* | *Chamaeangis sp* | 3 |
| | *Polystachya confusa* | 1 |
| | *Tridactyle sp* | 1 |
| *Tabernaemontana stapfiana* | *Bulbophyllum sp* | 3 |
| | *Polystachya confusa* | 4 |

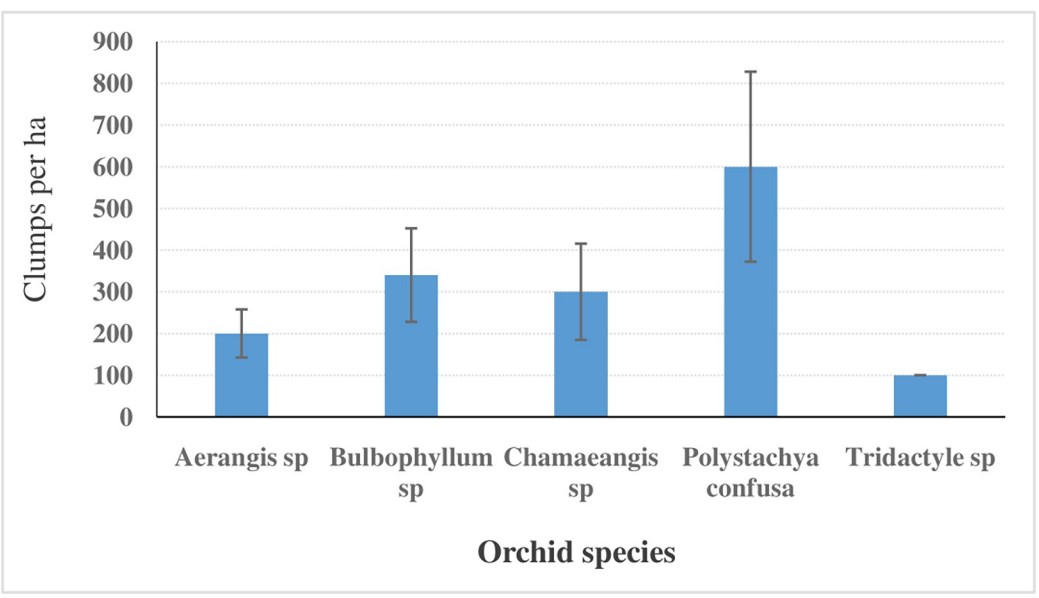

**Fig 3. Orchid species richness and respective population abundance in West Mau Forest, Kenya.**

any orchids (Table 4). The moderately disturbed forest had a higher total orchid population density (422.2±195.6 clumps per ha) than the intact forest (390±133.8 clumps per ha), but the difference was not statistically significant ($F_{(1,2)}$ = 0.01; $p$ = 0.933). Among individual orchid species, the population density of *Aerangis* sp was relatively higher (233.3±66.7 clumps per ha) in the moderately disturbed forest than in the intact forest (100.0 ±0.0 clumps per ha).

## 3.3 Host tree structural characteristics and orchid clump population

**3.3.1 Variation in orchid clump density with increase in host tree DBH size-class.** Host trees of smaller DBH (trees of 0.1–20.0 cm DBH) had between 700 and 1,700 clumps per ha (Fig 4). Medium size trees (20.1–40.0 cm DBH) had the largest number of orchid clumps at between 2,300 and 4,100 clumps per ha. There was a general reduction in orchid clumps to between 500 and 100 clumps per ha with increase in stem DBH from 40.1–100.0 cm DBH.

**3.3.2 Variation in orchid clump density with increase in height of host trees.** Relatively shorter host trees (6 m—20 m) had an average of 2,500 orchid clumps ha$^{-1}$ (Fig 5). Taller trees (20.1 m—30.0 m) had the largest orchid clump population density (5,700 clumps ha$^{-1}$). The tallest trees in the forest ecosystem (30.1–40 m in height) had only 900 clumps ha$^{-1}$ (Fig 5).

**Table 4. Orchid species richness and abundance across a forest degradation gradient in West Mau Forest Block, Kenya.**

| Forest disturbance regime | Orchid species | Abundance (clumps ha$^{-1}$) |
|---|---|---|
| Intact forest | *Aerangis sp* | 100.0 ±0.0 |
| | *Bulbophyllum sp* | 340.0 ±112.2 |
| | *Chamaeangis sp* | 300.0 ±115.5 |
| | *Polystachya confusa* | 671.4±360.4 |
| | *Tridactyle sp* | 100.0 ±0.0 |
| Moderately disturbed forest | *Aerangis sp* | 233.3±66.7 |
| | *Polystachya confusa* | 516.7±292.6 |
| Highly degraded forest | - | - |

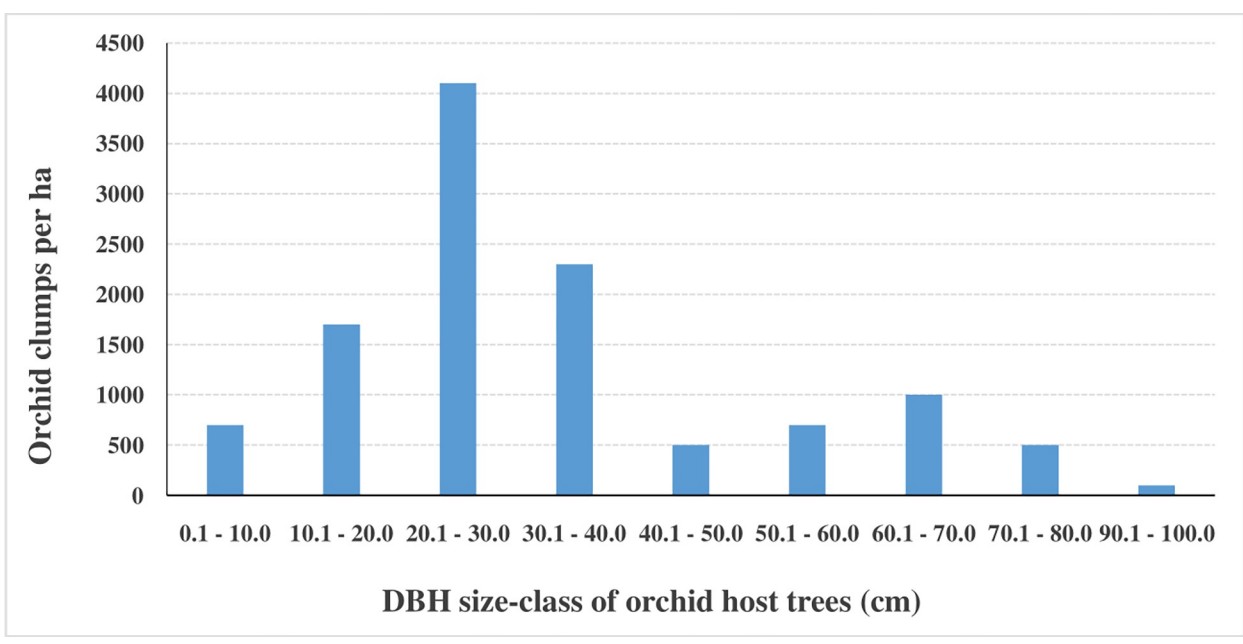

**Fig 4. Variation in orchid clump density with increase in host tree stem DBH.**

**3.3.3 Host tree bark texture and orchid clump establishment.** An aspect that we did not set out to investigate, but we stumpled on was the observation that orchid host trees tended to have rough or fissured backs (Table 5). Orchids did not burrow into the bark of the host trees. Instead, they anchored on the host-tree trunchs by attaching to mycorrhizal fungi. Relatively younger host trees tended to have smooth barks, and hence lacked mycorrhizal attachment to support orchids. The mean stem DBH for host trees whose trunks were initially smooth, but became rough at maturity was 24.5 cm, while that of host trees whose stems were rough from an early age was 12.1 cm (Table 5).

The occurrence of orchids was influenced host tree varied according to the tree height (Table 6). For example, *Albizia gummifera* and *Parvetta grandifolia* each had a single orchid clump located at a tree height of 21.7 m and 7 m respectively. The highest number of orchid clumps was observed in *Neoboutonia macrocalyx* with 11 clumps distributed at different heights ranging from 1.75 m to 13 m. *Syzygium guineense* had an orchid located at a height of 25 m compared to *Neoboutonia macrocalyx* that hosted an orchid at 1.75 m above the ground.

**Table 5. Bark texture of orchid host trees and respective orchid clumps.**

| Family | Host tree species | Bark texture | Mean stem DBH (cm) | Orchid clumps per tree | Orchid clumps ha$^{-1}$ |
|---|---|---|---|---|---|
| *Fabaceae* | *Albizia gummifera* | Rough when mature | 72.5 | 1 | 100 |
| *Flacourtiaceae* | *Dovyalis macrocalyx* | Rough | 12.1 | 3.8 | 633 |
| *Euphorbiaceae* | *Macaranga kilimandscharica* | Rough when mature | 51.7 | 1.8 | 200 |
| *Euphorbiaceae* | *Neoboutonia macrocalyx* | Rough when mature | 36.7 | 3.2 | 514 |
| *Rubiaceae* | *Parvetta grandifolia* | Fissured | 38.2 | 2.4 | 240 |
| *Podocarpaceae* | *Podocarpus sp* | Fissured when mature | 24.5 | 2 | 200 |
| *Araliaceae* | *Schefflera abyssinica* | Fissured when mature | 43.5 | 2 | 200 |
| *Myrtaceae* | *Syzygium guineense* | Rough | 70 | 1.7 | 167 |
| *Apocynaceae* | *Tabernaemontana stapfiana* | Rough | 36.5 | 2.7 | 270 |

**Table 6. Vertical height of each orchid clump on the host tree.**

| Host tree | Tree height (m) | Orchid clump height on host tree (m) | | | | | | | | | | |
|---|---|---|---|---|---|---|---|---|---|---|---|---|
| | | C1 | C2 | C3 | C4 | C5 | C6 | C7 | C8 | C9 | C10 | C11 |
| *Albizia gummifera* | 36 | 21.7 | | | | | | | | | | |
| *Dovyalis macrocalyx* | 6.6 | 2.4 | 3.6 | 3.5 | 3.5 | 4 | | | | | | |
| *Macaranga kilimandscharica* | 24.8 | 7 | 10 | 11.6 | 15 | 20 | | | | | | |
| *Neoboutonia macrocalyx* | 20.7 | 1.75 | 3.5 | 4.3 | 4.8 | 5.3 | 7.7 | 7.8 | 8.5 | 9 | 11.9 | 13 |
| *Parvetta grandifolia* | 15.8 | 4.38 | 4.5 | 5 | 7 | 8.6 | | | | | | |
| *Podocarpus sp* | 26 | 7 | | | | | | | | | | |
| *Schefflera abyssinica* | 18 | 10 | 13 | | | | | | | | | |
| *Syzygium guineense* | 34 | 22.7 | 25 | | | | | | | | | |
| *Tabernae montanastapfiana* | 18.8 | 4.1 | 5.5 | 5.6 | 5.7 | 6.6 | 7.1 | | | | | |

# 4. Discussion

## 4.1 Orchid species richness and abundance

This study showed that orchid species richness decreased with increase in the severity of forest degradation. The intact forest was significantly richer in orchid species than the moderately disturbed forest. The highly degraded forest had no orchids at all. This could be attributed to

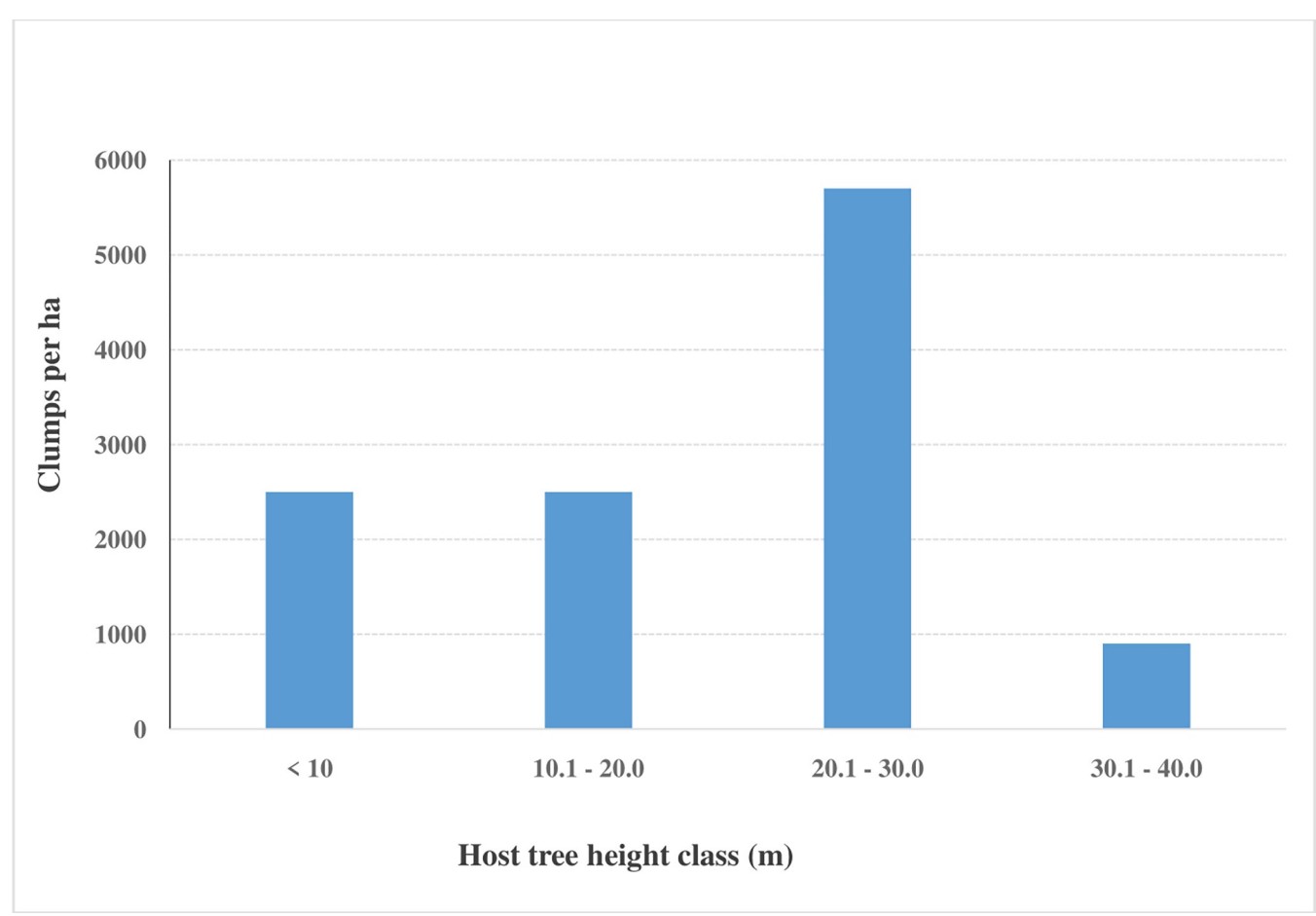

**Fig 5. Variation in orchid clump density with increase in the height of host trees.**

the presence of diverse host trees, a more complex canopy structure and a stable microclimate in the intact forest, which offered a favourable habitat for orchids. A similar observation was made by Besi, Mustafa [36] who indicated that intact forests have well-established canopy structures, balanced nutrient cycles and minimal human disturbance, which support diverse indigenopus trees that host orchids. The lower orchid species richness observed in the moderately disturbed forest is attributable majorly to human activities such as logging which alter the ecological stability suitable for the establishment and growth of orchids [37, 38]. It was also observed that certain orchids have specific mycorrhizal associations that are crucial for their germination, establishment and growth. Forest degradation easily disrupts these associations, impacting negatively on the ability of such orchids to establish themselves [39, 40]. The observation that highly degraded forest sites did not harbor any orchid species demonstrates the susceptibility of orchids to habitat disturbance and their likely role as bioindicators of ecosystem health. In this regard, several studies have described orchids as keystone plant species for monitoring ecosystem health and function [8, 41, 42].

Unlike orchid species richness, orchid population abundance illustrated a slightly different narrative. The intact forest had relatively fewer orchid clumps per ha than the moderately disturbed forest. This finding is similar to a study conducted in a disturbed tropical rainforest in Terengganu and Kelantan, Malaysia [19] which established that orchid clumps in moderately disturbed forest areas were more abundant compared to relatively intact forest areas and extremely degraded forest sites. The difference in the population of orchids across the forest degradation gradient is primarily a result of the availability of suitable microhabitats and canopy gaps, which influence the establishment of orchid clumps [23]. Essentially, certain amount of light is necessary for orchids to grow and blossom at their best. Thus, the degree of canopy cover, as well as canopy gaps attributable to forest degradation status, have an impact on the level of light availability within a forest stand. The understanding is that light that is diffused via the canopy may be more advantageous for orchid clump establishment than lack of light due to complete canopy closure or total exposure to sunlight caused by lack of a forest canopy [43]. However, moderate forest disturbance is a relative term. Minimal disturbance to the forest canopy tends to facilitate orchid clump development by providing access to sunlight and a suitable microclimate for their establishment and survival. Significant disturbance to the forest canopy, on the other hand, increases light intensity and alters the microclimate in such a way that damages orchid habitats.

### 4.2 Structural characteristics of host tree species

This study shows that approximately 28% of trees in West Mau Forest hosted orchid species. One key feature of these host tree species is that they tended to have rough or fissured barks, particularly when mature compared to most non-host trees with smooth barks. This observation is similar to that of Hernández-Pérez, Solano [44], who reported that the texture of the host tree bark is a critical determinant of the orchid species abundance and richness. In their study, fissured-bark trees hosted more orchids species (12 species), compared to smooth barks (5 species) and peeling barks (3 species). Rough or fissured barks host more orchid species and populations because they support the establishmment of mycorrhizal fungi, which orchids depended on for vital nutrients. In this regard, orchid species-host tree-specificity varies with the mycorrhizal relationship preferred by an orchid species.

Our results indicate that host trees of smaller DBH sizes (0.1–20.0 cm DBH) had relatively fewer orchid clumps (700 to 1,700 clumps per ha) than host trees of medium DBH sizes (20.1–40.0 cm DBH), which had the largest number of orchid clumps (2,300 to 4,100 clumps per ha). Host trees with the largest DBH sizes (40.1–100.0 cm DBH) had the least number of orchid

clumps (900 clumps per ha). Similar findings were made by Borrero, Alvarez [45] who reported that orchids preferred host trees with an average DBH of 31.5 cm. Adhikari, Fischer [23] also reported that the DBH sizes of host trees had a significant effect on the density of orchids clumps. This phenomenon can be attributed to the fact that smaller DBH size orchid host trees harbour fewer orchid clumps because their barks tend to be smooth at a younger age. As the trees mature and their barks become rough and fissured, they attract a significantly larger population of orchids [46]. As the trees grow larger and begin to age, their barks begin to peel off and this leads to the shedding of orchid clumps hence the reduction in orchid population among some of the trees with the largest DBH.

Our study showed that relatively shorter host trees (average height of 6 to 20 m) had fewer orchid clumps (2,500 clumps per ha) than relatively taller trees (average height of 20.1 to 30.0 m) with 5,700 clumps per ha, while the tallest trees (average height of 30.1 to 40.0 m) had the least orchid population (900 clumps per ha). Epiphytic orchids are able to utilize the limited resources available in the higher zone of host trees by by getting nutrients from decaying detritus and stem flow [47, 48]. This observation ties pretty well with earlier studies on tree DBH sizes because tree size increases synergistically in both stem diameter and crown height [49, 50]. Thus, as we attribute orchid species richness and population density to tree bark characteristics, it is also important to consider the contribution of gap dynamics. As illustrated by Jacquemyn, Micheneau [51], the spatial location of a tree within a forest ecosystem influences the microclimate and temperature conditions. In the case of West Mau Forest, shorter trees occupied the understory layer of the forest where light capture was limited. Orchids that established on these trees struggled to access light hence their relatively lower population density. Host trees of medium height mostly occupied the sub-canopy and main canopy layers of the forest. This is the zone of optimal conditions in terms of light access, microclimate and temperature hence the high orchid population density [19]. The tallest trees occupied the emergent layer of the forest where they were constantly exposed to direct sunlight, winds and frost. These conditions may not have been ideal for the symbiotic relationships that orchids depend on for their growth and survival hence the extremely low orchid population density [52].

## 5. Conclusion

This study demonstrated that orchid species richness decreased with increase in the severity of forest degradation. Intact forest blocks had significantly more orchid species compared to the moderately disturbed forest blocks. Highly degraded forest sites had no epiphytic orchards at all. The observation indicates that orchids can serve as good bioindicators of forest ecosystem health. It was also observed, however, that moderately disturbed forest blocks had higher orchid population density than intact forest blocks. The results of this study suggest that future forest logging operations should consider the value of forests as habitats for non-wood plant resources, such as orchids.

## Supporting information

**S1 Data.**
(XLS)

## Acknowledgments

The authors would like to acknowledge the Kenya Forest Service (KFS), West Mau region for allowing us access into the forest and their staff for providing guidance through the forest during the study period.

## Author Contributions

**Conceptualization:** Job N. Mirioba, William Emitaro, Benson Obwanga, Humphrey Gaya, Nereoh Leley, John Otuoma, Fanuel Kawaka.

**Formal analysis:** Job N. Mirioba, Nereoh Leley, John Otuoma.

**Funding acquisition:** John Otuoma, Fanuel Kawaka.

**Methodology:** Job N. Mirioba, William Emitaro, Benson Obwanga, Humphrey Gaya, Nereoh Leley, John Otuoma, John M. Maina, Fanuel Kawaka.

**Supervision:** William Emitaro, John Otuoma, John M. Maina, Fanuel Kawaka.

**Writing – original draft:** Job N. Mirioba, John M. Maina.

**Writing – review & editing:** John Otuoma, John M. Maina, Fanuel Kawaka.

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
