## [Decision Letter · Decision Letter 0]

29 Nov 2023

PONE-D-23-29375Orchid Species Diversity Across a Forest Disturbance Gradient in West Mau Forest, KenyaPLOS ONE

Dear Dr. Kawaka,

Thank you for submitting your manuscript to PLOS ONE. After careful consideration, we feel that it has merit but does not fully meet PLOS ONE’s publication criteria as it currently stands. Therefore, we invite you to submit a revised version of the manuscript that addresses the points raised during the review process.

**Please, improve your manuscript according to reviewer suggestions**

We look forward to receiving your revised manuscript.

Kind regards,

Marcela Pagano, Ph.D, M.D.

Academic Editor

PLOS ONE

Journal Requirements:

This study was supported by Rufford Foundation Completion Grant No.36278-C. We acknowledge the Kenya Forest Service (KFS), West Mau region for allowing us access into the forest and their staff for providing guidance through the forest.

Reviewers' comments:

Reviewer's Responses to Questions

**Comments to the Author**

1. Is the manuscript technically sound, and do the data support the conclusions?

Reviewer #1: Partly

2. Has the statistical analysis been performed appropriately and rigorously? 

Reviewer #1: I Don't Know

3. Have the authors made all data underlying the findings in their manuscript fully available?

Reviewer #1: No

4. Is the manuscript presented in an intelligible fashion and written in standard English?

Reviewer #1: Yes

5. Review Comments to the Author

Reviewer #1: The study analyses orchid diversity and abundance along a degradation gradient in West Mau Forest, Kenya.

Although I find the topic of interest and the paper reasonably well written, I have several major concerns.

Firstly, the methods section does not explain well the study design nor the performed analysis. The main setup of the study is comparing orchid diversity along a degradation gradient. However, no information is provided how the 3 sites representing this gradient were selected, how far apart they are, to what extent they differ in degradation level (how do they e.g. defer in forest structure?), if they are similar otherwise (e.g. same soil type, topography, etc.), no photos nor map is included, etc. Therefore, before it can be judged if the conclusions drawn from the analysis are correct, much more information about the study design is necessary.

Of what I do understand from the description of the scientific setup, only 1 site per degradation level is chosen if I understand correctly. This seems to me not enough. However, before this can be judged well, much more information about the current setup would be helpful.

Further, regarding the analysis itself, the description is also very short. It is for example not clear to me how orchid density per tree type was calculated. Why is the unit per ha and not per tree?

Another major concern that I have is that many statements are made in the discussion that are not backed up by data nor references (e.g. regarding the importance of rough bark for orchids). I recommend reformulating to make clear it concerns speculation, and/or back statements up by data that is presented in the results section.

Detailed comments:

In the study design section: It is not clear how many line transects were established per site nor how many subplots per plot nor how the number of 45 sample plots was reached.

In the data collection section: It is not clear to me at which level the data was collected. Was this per host tree?

In the data analysis section: It is not clear to me at which level species diversity and abundance was calculated, was this at the tree or plot level? Further, were any covariates taken into account in the statistical analysis? Any dependence structures tested?

The conclusion reads more as a recommendation than a conclusion.

6. PLOS authors have the option to publish the peer review history of their article (what does this mean?). If published, this will include your full peer review and any attached files.

Reviewer #1: No

---

## [Author Response · Author response to Decision Letter 0]

4 Jan 2024

4th January, 2024

Thanks for the insightful comments by the reviewers on our manuscript: Orchid Species Diversity across a Forest Disturbance Gradient in West Mau Forest, Kenya. 

As advised we have removed the funding information under acknowledgment in the manuscript. The new information under acknowledgement now reads as follows: The authors would like to acknowledge the Kenya Forest Service (KFS), West Mau region for allowing us access into the forest and their staff for providing guidance through the forest during the study period. 

In addition we have corrected the ‘Funding Information’ and ‘Financial Disclosure’ as suggested including correct grant numbers for the award. 

Regards,

Fanuel Kawaka

ON BEHALF OF THE AUTHORS

---

## [Decision Letter · Decision Letter 1]

31 May 2024

PONE-D-23-29375R1Orchid Species Diversity Across a Forest Disturbance Gradient in West Mau Forest, KenyaPLOS ONE

Dear Dr. Kawaka,

Thank you for submitting your manuscript to PLOS ONE. After careful consideration, we feel that it has merit but does not fully meet PLOS ONE’s publication criteria as it currently stands. Therefore, we invite you to submit a revised version of the manuscript that addresses the points raised during the review process.

We look forward to receiving your revised manuscript.

Kind regards,

RunGuo Zang

Academic Editor

PLOS ONE

Journal Requirements:

Additional Editor Comments:

Please respond to the new concerns of the referees

Reviewers' comments:

Reviewer's Responses to Questions

**Comments to the Author**

1. If the authors have adequately addressed your comments raised in a previous round of review and you feel that this manuscript is now acceptable for publication, you may indicate that here to bypass the “Comments to the Author” section, enter your conflict of interest statement in the “Confidential to Editor” section, and submit your "Accept" recommendation.

Reviewer #2: (No Response)

2. Is the manuscript technically sound, and do the data support the conclusions?

Reviewer #2: Partly

3. Has the statistical analysis been performed appropriately and rigorously? 

Reviewer #2: Yes

4. Have the authors made all data underlying the findings in their manuscript fully available?

Reviewer #2: Yes

5. Is the manuscript presented in an intelligible fashion and written in standard English?

Reviewer #2: Yes

6. Review Comments to the Author

Reviewer #2: The revised manuscript titled “Orchid Species Diversity Across a Forest Disturbance Gradient in West Mau Forest, Kenya” provides some interesting data about the influence of forest degradation on orchid species richness and abundance. Although the study design, analyses and use of statistics generally seem to be well developed, there are several points throughout the manuscript that remain unclear, which should be revised and explained/discussed in more detail (see comments made within the document). For example, in the methods section it would be important to present a table, including a classification of the three different forest disturbance regimes. Also, it is necessary to describe the sampling and identification of orchids in more detail, and it would be helpful to present a photo plate of the species and a map of the study area. In the results section should be mentioned the growth form (epiphytic or terrestrial) of the orchids and also the family and bark type of each host tree. Furthermore, it would be important to analyze the available data on the height of each clump on the host tree or within vertical zones to see if there are differences between orchid species. These ecological aspects need to be discussed in more detail. In the introduction and discussion, the authors demonstrate good knowledge of the necessary background literature, although some relevant studies should be still considered to strengthen the content and frame the paper in a wider context. Besides, many concluding statements should be supported by references.

7. PLOS authors have the option to publish the peer review history of their article (what does this mean?). If published, this will include your full peer review and any attached files.

Reviewer #2: No

---

## [Author Response · Author response to Decision Letter 1]

10 Jul 2024

28th June, 2024

Thanks for the insightful comments by the reviewers on our manuscript: Orchid Species Diversity across a Forest Disturbance Gradient in West Mau Forest, Kenya. We have used the comments to revise and improve our manuscript. We have provided point by point responses to the concerns raised by the reviewers on our rebuttal outlined in the author response to the reviewer comments. 

We hope that the revised manuscript findings will be of interest and beneficial to the readers of PLOS One. On behalf of all the authors, I further confirm that this manuscript is original, has not been published before and is not currently being considered for publication elsewhere and wish to confirm that authors declare that there is no known conflict of interests regarding the publication of this article. In addition, the manuscript has been read and approved by all the named authors and that there are no other persons who satisfied the criteria for authorship but are not listed. The authors understand that the corresponding author is the sole contact for the editorial process and will be responsible for communicating with the other authors about progress, submissions of revisions and final approval of proofs.

Regards,

Fanuel Kawaka

ON BEHALF OF THE AUTHORS

---

## [Editor Report · Decision Letter 2]

15 Jul 2024

Orchid Species Diversity Across a Forest Disturbance Gradient in West Mau Forest, Kenya

PONE-D-23-29375R2

Dear Dr. Kawaka,

We’re pleased to inform you that your manuscript has been judged scientifically suitable for publication and will be formally accepted for publication once it meets all outstanding technical requirements.

Kind regards,

RunGuo Zang

Academic Editor

PLOS ONE
---

## [Editor Report · Acceptance letter]

26 Jul 2024

PONE-D-23-29375R2 

PLOS ONE

Dear Dr. Kawaka, 

I'm pleased to inform you that your manuscript has been deemed suitable for publication in PLOS ONE. Congratulations! Your manuscript is now being handed over to our production team.

Kind regards, 

on behalf of

Professor RunGuo Zang 

Academic Editor

PLOS ONE